# PET Radiopharmaceuticals for Specific Bacteria Imaging: A Systematic Review

**DOI:** 10.3390/jcm8020197

**Published:** 2019-02-06

**Authors:** Sveva Auletta, Michela Varani, Rika Horvat, Filippo Galli, Alberto Signore, Søren Hess

**Affiliations:** 1Nuclear Medicine Unit, Department of Medical-Surgical Sciences and of Translational Medicine, “Sapienza” University of Rome, 00161 Rome, Italy; sveva.auletta@hotmail.it (S.A.); varanimichela@gmail.com (M.V.); filippo.galli@hotmail.com (F.G.); 2Department of Radiology and Nuclear Medicine, Hospital of Southwest Jutland, 6700 Esbjerg, Denmark; Rika.Horvat@rsyd.dk (R.H.); soren.hess@rsyd.dk (S.H.); 3Department of Regional Health Research, Faculty of Health Sciences, University of Southern Denmark, 5230 Odense, Denmark

**Keywords:** PET, bacteria, nuclear medicine imaging, infection

## Abstract

Background: Bacterial infections are still one of the main factors associated with mortality worldwide. Many radiopharmaceuticals were developed for bacterial imaging, both with single photon emission computed tomography (SPECT) and positron emission tomography (PET) isotopes. This review focuses on PET radiopharmaceuticals, performing a systematic literature review of published studies between 2005 and 2018. Methods: A systematic review of published studies between 2005 and 2018 was performed. A team of reviewers independently screened for eligible studies. Because of differences between studies, we pooled the data where possible, otherwise, we described separately. Quality of evidence was assessed by *Quality Assessment of Diagnostic Accuracy Studies* (QUADAS) approach. Results: Eligible papers included 35 published studies. Because of the heterogeneity of animal models and bacterial strains, we classified studies in relation to the type of bacterium: Gram-positive, Gram-negative, Gram-positive and negative, others. Conclusions: Results highlighted the availability of many promising PET radiopharmaceuticals for bacterial imaging, despite some bias related to animal selection and index test, but few have been translated to human subjects. Results showed a lack of standardized infection models and experimental settings.

## 1. Introduction

For several decades, nuclear medicine modalities like three-phase bone scintigraphy, ^67^Ga-citrate scintigraphy, and radiolabeled white blood cell scintigraphy have been used for imaging infection and inflammation. In the last years, new techniques and new radiopharmaceuticals have been proposed such as 2-deoxy-2[^18^F]fluoro-D-glucose ([^18^F]FDG) [1,2] and many bacterial specific agents [3]. 

Early in the history of clinical whole-body [^18^F]FDG-PET occasional reports of false positive findings due to inflammation in patients with malignant tumors were considered a nuisance caused by the non-specificity of [^18^F]FDG [4]. Tahara et al. were among the firsts to report specific [^18^F]FDG accumulation in abdominal abscesses [5]. It is now well-known that [^18^F]FDG is taken up by cells involved in the inflammatory response (e.g., neutrophils, macrophages and activated leukocytes) because they express high levels of glucose transporters like malignant cells (albeit not to the same extent), and in addition, circulating cytokines seem to increase the affinity of these glucose transporters for [^18^F]FDG [6]. Moreover, [^18^F]FDG uptake has been shown to correlate very well with the inflamed tissue in both septic and aseptic inflammation [7,8] and it has recently been shown that some bacterial strain can also bind [^18^F]FDG [9]. 

Nevertheless, the non-specificity is now considered an asset in some settings, e.g., fever of unknown origin, a heterogeneous group of diseases with a multitude of differential diagnoses of infectious, malignant, or inflammatory etiology most of which have an element of hypermetabolism. As a sensitive whole-body modality, [^18^F]FDG-PET/CT may guide the clinicians towards more specific investigations to secure an etiologic diagnosis, and many infectious and inflammatory diseases have a systemic component to merit [^18^F]FDG whole-body imaging [2]. 

Nonetheless, the non-specific nature of [^18^F]FDG may also be a hindrance in other settings, as the distinction between aseptic inflammation and infectious foci is difficult. Much work has been invested in optimizing the use of [^18^F]FDG and much still lies ahead, e.g., better interpretation criteria, more elaborate quantitative techniques, and multiple-time-point imaging, which all have shown considerable promise in some settings [10,11]. However, just as tumour imaging has seen the introduction of several more specific PET-radiopharmaceuticals with alternative isotopes (e.g., prostate-specific membrane antigen (PSMA)-imaging for prostate cancer and various radiopharmaceuticals for neuroendocrine tumours), more specific radiopharmaceuticals are also being investigated for infection and inflammation. We have previously reviewed available radiopharmaceuticals for gamma camera imaging [3,12], but also for PET imaging, and a multitude of alternative candidates have been presented and assessed pre-clinically. Unfortunately, only few have been translated to humans with limited success. 

Early on, various broad-spectrum fluorine-18-based radiopharmaceuticals known from cancer imaging were explored, perhaps not surprising as they were readily available and because they aimed at for instance increased proliferation which also applies to bacteria, e.g., 3′-deoxy-3′-[^18^F]fluorothymidine ([^18^F]FLT) in *S. aureus*, but results did not transcend the non-specificity of [^18^F]FDG [13,14,15]. Similarly, with the availability of gallium-68, a PET-equivalent of ^67^Ga-citrate and other radiopharmaceuticals like [^68^Ga]gallium-UBI were developed and tested in animal models and humans. The use of gallium-68 has many advantages, including a favourable half-life of 68 min and generator availability, but despite promising results they were not convincingly better than [^18^F]FDG [16,17,18].

Other isotopes have been also tested, e.g., copper-64 (^64^Cu) bound to a peptide targeting a leucocyte receptor, but not directed specifically against a microorganism [19]. Finally, it is worth mentioning that white blood cells (WBC) labelling with [^18^F]FDG has being explored, but although initial studies were very promising, later results were more equivocal [20,21,22,23], thus techenetium-99m and indium-111-labelled WBC remain the gold standard technique for imaging infections [24].

Thus, in recent years, interest has turned towards more infection-specific radiopharmaceuticals, i.e., radiopharmaceuticals aimed directly at the microorganism and not only at the secondary inflammatory response [25]. The ability to directly diagnose specific microorganisms would be very attractive, especially in areas not readily accessible for sampling. In an attempt to qualify the discussion, we aimed to systematically survey and review the literature to evaluate the current status of the many potential candidates that have been presented. 

## 2. Materials and Methods

### 2.1. Inclusion Criteria

In the present review, only published articles that met the following criteria were included: articles (only original papers); radiopharmaceutical (any non-[^18^F]FDG-PET radiopharmaceutical); target (only specific for bacteria); experimental design (only in vivo targeting studies). Publications selected for this review were from peer-reviewed indexed journals.

### 2.2. Exclusion Criteria

Reviews, case reports, abstracts, editorials, poster presentations, publications in languages other than English, studies regarding [^18^F]FDG, studies in which compounds were not essentially specific for bacteria, articles with only in vitro studies or technical data, were excluded. The decision to include or exclude an article was made by consensus.

### 2.3. Search Methods for Identificationof Studies

The selection of published articles was based on the PRISMA guidelines [26].

We selected all studies published in English language regardless of publication status (published, ahead of print, in press, etc.). The following algorithm was used: (PET AND bacteria*) OR (PET and infect*); filters: published in the last 10 years. We searched for published papers present in PubMed (last 10 years) and Scopus (2005–2018). The literature search was broadened to all reference lists of all retrieved.

### 2.4. Data Extraction and Management

All authors screened independently full-text manuscripts for eligibility, reporting essential data in a summary table. Risk assessment of any potential bias and data collection was performed by using a standardised questionnaire, adjusted to included studies. Any disagreement was resolved through discussion.

### 2.5. Risk Assessment of Bias in Included Studies

The included studies were evaluated by using the *Quality Assessment of Diagnostic Accuracy Studies* (QUADAS) approach for any potential bias and variation [27]. We analysed the following variables: (1). animal selection bias (animal origin, animal model); (2). infection model bias (bacterium origin, bacterium number); (3). experimental design variation (infection model, radiopharmaceutical purity and specific activity); (4). reference standard bias (inappropriate reference standard, incorporation bias); (5). reference standard variation (definition of control model); (6). flow and timing bias (infection progression bias, radiopharmaceutical administration time, imaging time bias, uninterpretable test results, sample size). Each study was analyzed providing this information. Considering the heterogeneity of animal models and bacterial strains, we classified studies in relation to the type of bacterium (Gram-positive, Gram-negative, Gram-positive and negative, others).

## 3. Results

### 3.1. Data Synthesis 

Because of differences between studies (bacterial strain, infection model, isotope, compound, reference standard), data were pooled, when possible. When not, they were separately described. We found 69 potential studies that evaluated PET radiopharmaceuticals for bacterial imaging. After excluding the non-eligible articles, we analysed 35 published studies, shown in Figure 1. A summary of results is reported in Table 1, whereas a summary of QUADAS analysis is reported in Table 2.

### 3.2. Gram-Positive

We included 15 studies for the analysis, excluding those that did not meet the inclusion criteria. In total, ten radiopharmaceuticals were considered in the QUADAS analysis.

#### 3.2.1. Fluorine-18 (^18^F)-Labelled Radiopharmaceuticals

For many and well-known reasons, fluorine-18 was the first PET radioisotope of choice, i.e., a ubiquitous isotope with suitable half-life and well-established chemical properties. Thus, several potentially bacteria-specific radiopharmaceuticals labelled with fluorine-18 have been explored. 

Mills et al. looked at the alternative glucose analogue, [^18^F]FDG-6-phoshate, as a potential substrate for the bacterial universal hexose phosphate transporter (UHPT). UHPT is expressed in many bacteria species and provides the mechanism for sugar phosphate internalization in staphylococci in contrast to mammalian cells that are not able to directly transport sugar into the cell. The authors corroborated this premise in vitro by demonstrating very limited uptake of [^18^F]FDG-6-phosphate in UHPT-deficient *S. aureus* and significantly less than regular strains. Uptake in various mammalian cell lines was limited to the same extent as in UHPT-deficient *S. aureus* and significantly different from *S. aureus*. However, these results did not translate in vivo in a mouse infection model when comparing uptake of [^18^F]FDG-6-phosphate to conventional [^18^F]FDG at indwelling catheters with or without inoculation with *S. aureus*. The authors surprisingly observed a biodistribution similar to [^18^F]FDG both visually and semi-quantitatively, and they established that this was not due to dephosphorylation of [^18^F]FDG-6-phosphate in the blood to yield [^18^F]FDG. Furthermore, in contrast with [^18^F]FDG, there was no accumulation of [^18^F]FDG-6-phosphate in sterile inflammation at the tip of the indwelling catheters in non-infected mice, although the [^18^F]FDG uptake was higher in infected areas compared to sterile inflammation [46]. According to the authors, this study suggests that the majority of radiopharmaceuticals never reaches the bacteria. The lack of bacterial origin could have introduced some bias in the study.

Takemiya et al. looked at another modified pentasaccharide, ^18^F-labelled fluoromaltohexaose (FMH), targeting the maltodextrin transporter which has previously been explored in Gram-negative bacteria with comparable probes (see below) [31]. The target here was chosen because the uptake rates are high and the transporter is selectively expressed in bacteria. Furthermore, the substrates have limited toxicity to humans and the hydrophilic nature grants higher rates of clearance from non-infected tissues [63,64]. The authors investigated an infrared dye probe (MDP) detectable through the skin aimed at superficial (pocket) infections and the radiolabelled fluoromaltohexaose for deep infections. They compared both to each other and the latter also to [^18^F]FDG imaging in a rat model comparing microbially infected mock-up implants to sterile, turpentine-induced inflammation, and controls with no additional intervention besides the mock-up implants. The authors found corresponding visually increased uptake of MDP and FMH around the infected implants compared to both sterile inflammation and controls. For FMH, SUV_mean_ and SUV_max_ were also significantly higher in the infected rats compared to sterile inflammation and controls. Not surprisingly, based on pre-clinical and clinical experience, [^18^F]FDG uptake around infected implants and in the controls corresponded to FMH but was also increased at sites of sterile inflammation. Thus, the authors found FMH a potential probe for specific bacterial imaging with the ability to differentiate sterile inflammation from infection. However, the authors pointed out the caveat of translation comparing their results to the successful pre-clinical results from Zhang et al. that were not confirmed when translated to humans [40]. The non-specified origin of bacteria, the unclear infection model used and imaging time might have introduced risks of bias in the QUADAS analysis.

Langer et al. investigated ^18^F-labelled ciprofloxacin, an antibiotic previously labelled with techenetium-99m leading to discordant results. The alleged bacteria specific binding site is the topoisomerase II enzyme, the therapeutic target for ciprofloxacin. The authors performed in vitro pharmacokinetic studies and studied basic uptake mechanisms in vitro, albeit with Gram-negative *E. coli*. They concluded that no apparent specific uptake mechanism was present – increased uptake of [^18^F]ciprofloxacin was rapidly cleared, and was unaffected by large amounts of unlabelled radiopharmaceutical, as also reported by others using [^99m^Tc]technetium-ciprofloxacin [65]. Thus, the authors concluded that increased uptake was most likely due to non-specific physiologic increase in blood flow as part of the inflammatory response. Similar results were reached in four patients with Gram-positive soft tissue infection caused by ciprofloxacin sensitive microbes. Thus, although there was rapid uptake in infected tissue different from non-infected tissue, the time-activity curves and the efflux of radiopharmaceutical were similar, and just as reversible as time-activity-curves of blood cells also indicating that non-specific processes such as blood flow is more likely to account for the difference in uptake than specific intracellular binding [59]. The unclear mechanism of patients’ selection could be source of bias as well as the non-specified reference standard and study flow.

Recently, Zhang and colleagues radiolabelled the para-aminobenzoic acid (PABA) with fluorine-18 with the aim to detect *S. aureus* infection rat model and monitor drug response. Results showed a rapid renal clearance and an evident localization of 2-[^18^F]F-PABA in the infectious lesions in comparison to sterile inflammation induced by using heat killed bacteria. Authors suggested that 2-[^18^F]F-PABA might be a novel radiopharmaceutical for rapid and non-invasive imaging of a wide range of infections [62]. No sources of bias were reported for this study.

In summary, two of the ^18^F-labelled radiopharmaceuticals were able to discriminate the presence of bacteria from sterile inflammation or healthy control and, when compared to [^18^F]FDG, they showed better results. However, poor specificity was observed for [^18^F]ciprofloxacin in vitro and in vivo, and the authors questioned the presence of a bacteria-specific uptake mechanism. In the QUADAS analysis, the main source of bias was related to unspecified origins of animals or bacteria.

#### 3.2.2. Gallium-68 (^68^Ga)-Labelled Radiopharmaceuticals

The first study describing a ^68^Ga-radiopharmaceutical for imaging infections was published by Kumar et al. in 2011 and focused on the role of iron metabolism for bacterial growth [56]. It is well-known that tight regulation of iron availability is a key component in the human host response to bacterial invasion; concentrations of circulating free iron is too low to sustain bacterial growth so bacteria had to develop methods to acquire the necessary amounts from the abundance of iron bound to transport proteins like transferrin, e.g., siderophores or transferrin binding proteins like *S. aureus* does [66,67]. Since transferrin’s ability to bind Ga-ions is similar to its iron-binding ability, the exploration of a radioactive Ga-transferrin complex seemed an obvious choice as a potential bacteria specific radiopharmaceutical [56]. Infection was induced by intramuscular inoculation, and subsequent imaging with [^68^Ga]apo-transferrin showed focal time-dependent uptake at the site of infection. Increased uptake was also seen in one patient with an incidental wound infection by *Proteus mirabilis*. The authors also tested pure, unconjugated [^68^Ga]GaCl_3_ as control and no uptake was seen at the sites of infection. However, the paper states no information on reference standard for infection other than clinically established abscess at the site of inoculation which weakens the conclusion somewhat, introducing high risks of bias for QUADAS analysis. Another approach with gallium-68 was explored by two different groups using [^68^Ga]gallium-NOTA-UBI-29-41, a synthetic fragment derived from the antimicrobial peptide ubiquicidin that is part of the human first line response towards pathogens. The radiopharmaceutical had already been successfully explored labelled with techenetium-99m and was able to differentiate infection from sterile inflammation and tumours, both in vitro and in vivo, in both animals and humans [32,44,51,65], presumably the cationic antimicrobial peptide fragments interact electrostatically with anionic parts of the bacterial membrane [32,51]. 

In the first paper, Ebenhan et al. studied the normal biodistribution in rabbits. Furthermore, they found significantly increased uptake in muscular abscesses as compared to turpentine-induced sterile muscle inflammation, asthma-like lung inflammation and healthy controls. The target/non-target (T/NT) ratios were as high as 5-fold higher for infected muscle, whereas ratios between inflamed and normal muscle as wells as ratios between inflamed and normal lung tissue were almost similar [51]. The unclear origin of animals and used infection model might be sources of bias.

Vilche et al. used a similar setup, albeit with mice and heat shock-treated bacteria instead of turpentine for sterile inflammation, but their results were similar to those published by Ebenhan [44]. However, the non-specified origin of animals and unclear infection model used in the study might have introduced risks of bias. 

In a recent study, Bhatt et al. carried out several sub-studies on [^68^Ga]gallium-UBI-31-38 including in vitro tests for purity and stability and the assessment of uptake specificity in *S. aureus.* In vivo imaging was performed in an animal model of infection comparing the uptake in muscle inoculated with living *S. aureus* to muscle inoculated with heat-killed *S. aureus*. Finally, they undertook preliminary clinical studies in two patients and one control subject. In vitro they demonstrated a high and specific [^68^Ga]gallium-UBI-31-38 uptake in *S. aureus*. In preclinical studies they found a significantly higher uptake in infected muscle versus inflamed muscle with a T/NT-ratio of 3.24 ± 0.7. The clinical pilot study demonstrated high [^68^Ga]gallium-UBI-31-38 uptake in infectious foci in the lungs and diffusely around a knee prosthesis, both confirmed with biopsy, whereas no uptake was seen around a suspected hip prosthetic infection later confirmed to be non-septic inflammation. However, the reference standard for the preclinical trials are difficult to assess and the patient cases were highly selected and with very different infectious sites with no data presented on the underlying infections afterwards. The authors concluded that the method is promising, but still needs further validation [28].

In a similarly recent paper, Ebenhan et al. also performed a small first-in-man study with [^68^Ga]gallium-NOTA-UBI following successful in vitro assessment of cytotoxicity, and preclinical biodistribution and dosimetry studies in mice. In this study, [^68^Ga]gallium-NOTA-UBI uptake in knees was found in all three patients with suspected or known soft-tissue or bone infections, but not in the two healthy controls. No information is provided on the bacterial strains involved or if any bacteria was cultured at all, because all diagnoses of infection were based on other imaging modalities like bone scintigraphy and white-blood-cell scintigraphy. Authors also concluded that the method is promising, but with the need for further evaluation [32]. 

Nielsen et al. investigated the peptide A9 supposed to bind to *S. aureus* biofilm to establish its ability to distinguish infection from turpentine-induced sterile inflammation compared to [^18^F]FDG. Both radiopharmaceuticals showed increased uptake in both infection and inflammation with no discernable differences, although [^18^F]FDG was false negative in two cases. Furthermore, A9 uptake was higher in inflamed lesions as compared to infections. The authors speculated and then confirmed what the precise target of the A9 peptide was the cell membrane of dead bacteria [29]. The non-specified origin of bacteria and the unclear infection model used and study flow could be source of bias, resulting in applicability concerns too. 

Finally, Satpati et al. investigated ciprofloxacin once more, but labelled with gallium-68 with two different ligands. In vitro studies demonstrated only limited binding to bacterial cells, but when radiopharmaceutical uptake was compared in vivo in infected and turpentine induced inflamed soft tissue abscess there was increased uptake in both infection and inflammation, but more so in infection [58]. The unclear origin of animals might have introduced bias in the QUADAS analysis, leading to concerns about applicability too.

In summary, most ^68^Ga-labelled radiopharmaceuticals are promising and potential agents for distinguishing infection from sterile inflammation, but two studies were equivocal and further investigations were suggested [32,58]. The only radiopharmaceutical that was not shown to be selective for bacterial imaging was [^68^Ga]gallium-DOTA-A9 [29]. In the QUADAS analysis, the non-specified or unclear origin of animals and bacterial cells are the main causes of potential bias.

#### 3.2.3. Zirconium-89 (^89^Zr)-Labelled Radiopharmaceuticals

Pickett et al. looked specifically at *S. aureus*-infected prosthetic joint and assessed by a ^89^Zr-labelled monoclonal antibody (mAb) directed against the specific Gram-positive surface molecule lipoteichoic acid ([^89^Zr]zirconiumSAC55). Binding affinity was tested in vitro, and the diagnostic effectiveness for differentiating aseptic and septic prosthetic joint infection was tested in vivo in a mouse model and compared to [^18^F]FDG, [^18^F]NaF, and conventional CT with bioluminescent imaging as the reference standard. [^89^Zr]zirconiumSAC55 had significantly higher uptake in infected areas compared to sterile prostheses sites as well as control radiopharmaceutical without bacterial affinity. Similar to the human experience, [^18^F]FDG and [^18^F]NaF demonstrated increased uptake at infected sites as well as in areas of sterile postoperative inflammation, thus underlining radiopharmaceutical inability to distinguish active infection from background inflammation or slow indolent infections with low bacteria burden. Interestingly, zirconium-89 was chosen for its relatively long half-life (78 h) because of the long clearance time of mAbs necessitates delay between injection and imaging to optimize target-to-background ratio, but in this study the authors found that optimal imaging time was at 24 h post-injection [34]. No sources of bias were reported for this study. 

#### 3.2.4. Copper-64 (^64^Cu)-Labelled Radiopharmaceuticals

A special feature of *S. aureus* is its ability to initiate bacterial colonization under flow conditions at sites of endothelial damage. A key element, albeit not completely understood, is the effects of staphylocoagulase that enables *S. aureus* to clot blood and evade the host immune response. Panizzi et al. looked specifically at the role and expression of this enzyme by way of a prothrombin analogue able to form a SC-prothombin complex. For PET-imaging the prothrombin analogue was chelated with DTPA and subsequently labelled with copper-64 ([^64^Cu]copper-DTPA-ProT). The authors found a robust accumulation of the radiopharmaceutical at infected vegetation in a mouse endocarditis model, whereas control mice with endothelial damage but no infection demonstrated no uptake of [^64^Cu]copper-DTPA-ProT. The presence of infection was corroborated by bioluminescent *S. aureus* strains which correlated well with radiopharmaceutical uptake both in vivo and at ex vivo autoradiography of the infection targets in the aortic root. Additional in vivo experiments also substantiated the findings; a substantially increased accumulation was seen in *S. aureus* compared to mice without bacteraemia and mice infected with coagulase negative *S. epidermidis*. Similarly, no uptake was present in mice infected with coagulase deficient *S. aureus* strains [54]. For this study, the QUADAS analysis reported some bias concerning the unclear origin of animals, the methodology of experiments and study flow. 

#### 3.2.5. Iodine-124 (^124^I)-Labelled Radiopharmaceuticals

The nucleoside analogue 1-2′-deoxy-2′-fluoro-beta-D-arabinofuranosyl-5-iodouracil (FIAU) was initially explored as an anti-viral drug, but was discontinued due to severe toxicity. Later it was discovered serendipitously to be a substrate for thymidine kinases in several bacteria; it enters cells freely and become trapped inside the cells after phosphorylation and, as a diagnostic agent, the required dose is well below the limit of toxicity. Furthermore, it is not a significant substrate for the most abundant human thymidine kinases thereby making it an interesting radiopharmaceutical for infection imaging. The choice of a long-lived radioisotope as iodine-124 with a half-life of 4.2 days was based on practical considerations (better production and transport logistics and better chemical yield than fluorine-18) as well as clinical considerations (the possibility of late imaging, as chronic prosthetic joint infections may involve relatively slow-growing bacteria) [40,68].

Diaz et al. examined eight patients, and one healthy control. Scans were positive in 7/8 patients with positive culture in 6/7 and negative culture in one patient who was however deemed positive for infection nonetheless due to clinical history and findings. Scan was negative in one patient with resolution of symptoms without surgery or antibiotics and in the healthy control subject. Thus, the results seemed promising, but the whole-body distribution showed significant uptake in liver, kidneys, muscles and to a lesser extent in several other organs, due in part to the presence of a mitochondrial enzyme resembling the bacterial TK. This may hamper the general applicability in suspected infections in these organs. The target TK is considered specific for bacteria and therefore potentially able to differentiate infection from non-infectious inflammation. This was supported as the healthy control had osteoarthritis in a knee which showed no uptake, but, on the other hand, in the one patient with positive uptake but negative culture, the biopsy showed chronic inflammation. This was attributed to infection, but it may be that non-infected inflammation was the cause of uptake which would also hamper the clinical use of this radiopharmaceutical [57]. It is also worth noting that patients were probably highly selected, albeit no data is presented on patient selection, leading to high risk of bias in the QUADAS analysis.

Zhang et al. assessed the same radiopharmaceutical in the setting of suspected prosthetic joint infection (PJI) [40]. A preliminary phase I study in 6 patients and 6 healthy controls established its biodistribution, dosimetry and safety, but the phase II study did not confirm the promising results of the abovementioned study by Diaz et al. Indeed, in 19 patients suspected of PJI, four were treated as positive with final diagnosis determined by an independent adjudication board with access to all information. However, increased background uptake in muscles and artificially increased uptake around the prosthesis due to metal artefacts and attenuation obscured any pathologic activity and rendered the differentiation between infected and non-infected prosthesis impossible. Various semi-quantitative approaches (e.g., TBR and SUV_max_) were also unsuccessful. Thus, the authors concluded that while well-tolerated and with acceptable dosimetry, the clinical applicability seems limited due to poor image quality and low specificity, at least in suspected PJI or other settings with metal artefacts or very low bacterial burden. The unclear patients’ selection process and the not-well specified index test and reference standard might have led to the presence of bias in the QUADAS analysis.

### 3.3. Gram-Negative

We included 10 studies for the analysis, excluding those that did not meet the inclusion criteria. In total, 8 radiopharmaceuticals and 4 different bacterial strains were considered in the QUADAS analysis.

#### 3.3.1. Carbon-11 (^11^C)-Labelled Radiopharmaceuticals

Mutch and colleagues radiolabelled PABA with carbon-11 to study the targeting of bacterial metabolism in *E. coli* mice infection model, using heat-killed bacteria as control. They showed a significant uptake of radiopharmaceutical in the infectious site compared to inflamed foci, concluding that [^11^C]carbonPABA is an attractive candidate to image living bacteria in humans [30]. No potential source of bias was reported for this study. 

#### 3.3.2. Fluorine-18 (^18^F)-Labelled Radiopharmaceuticals

As aforementioned, fluorine-18 is one of the most studied PET isotopes and many compounds radiolabelled with ^18^F have been explored for imaging of Gram-negative bacteria.

Yao et al. compared the biodistribution and specificity of ^18^F-2-fluorodeoxysorbitol ([^18^F]FDS) to [^18^F]FDG in a *E. coli* mouse infection model and in humans. They concluded that [^18^F]FDS is a potential imaging agent for the diagnosis of Enterobacteriaceae infections in preclinical setting. Given the lack of adverse events and rapid clearance through the urinary system, it could easily enter into clinical practice [42]. Potential bias may have been introduced by the fact that the origin of animals and bacterial cells was non-specified in the paper. Therefore, lack of reproducibility may also be a critical issue. The ability of [^18^F]FDS to discriminate septic from sterile inflammation has also been investigated by another group in several Enterobacteriaceae infection mouse models comprising *E. coli* infection, mixed infection with *E. coli* and *S. aureus*, *K. pneumoniae* infection and brain infection. Results showed that [^18^F]FDS was able to image infectious foci with higher specificity when compared to [^18^F]FDG. Therefore, it could be a candidate imaging radiopharmaceutical of Enterobacteriaceae infections to be translated into humans [48]. Also in this study, the lack of any information about animal origin could have introduced some biases. Moreover, the use of different SUV scale between [^18^F]FDS and [^18^F]FDG-PET acquisition could have been another source of bias in the QUADAS analysis. Recently, the selectivity and sensitivity of [^18^F]FDS have been also investigated in *K. pneumoniae* lung infection animal model by Li and colleagues. They compared this radiopharmaceutical with [^18^F]FDG in septic and aseptic inflammatory condition [33]. Authors reported higher specificity of [^18^F]FDS rather than [^18^F]FDG, not only in discriminating septic from sterile inflammation, but also Gram-negative from Gram-positive bacteria, proposing [^18^F]FDS as new clinical standard imaging agent for lung infection. No sources of bias were reported for this study.

Then, Ning et al. performed a study to evaluate the sensitivity and specificity of fluorine-18 radiolabelled maltohexaose ([^18^F]MH) that targets the bacteria-specific maltodextrin transporter, in *E. coli* infection preclinical model. They investigated both [^18^F]MH and [^18^F]FDG in several experimental conditions such as the use of different bacterial amounts, live and dead bacteria and drug-resistant *E. coli*. They demonstrated a better in vivo performance of [^18^F]MH than [^18^F]FDG for detection of infection at early stage and identification of drug resistance. Therefore, they concluded that it is a potential radiopharmaceutical to be introduced in clinical practice [49]. No potential sources of bias were reported for this study.

Recently, the 6′’-[^18^F]fluoromaltose was investigated as specific imaging radiopharmaceutical of bacterial infection by Gowrishankar et al. [50]. Results showed that 6′’-[^18^F]fluoromaltotriose is taken up by viable *E. coli* infected muscle rather than control infected with heat-inactivated bacteria, discriminating between infection and inflammation. The lack of animal origin could have introduced biases in the study, thus limiting its applicability.

Martìnez and colleagues developed the 2-deoxy-2-[^18^F]fluoroacetamido-D-glucopyranose ([^18^F]FAG) with the aim to evaluate its specificity for bacterial infection in *E. coli* infection rat model in comparison to [^18^F]FDG. They reported higher and significant uptake of [^18^F]FAG in infectious foci than inflammatory sites, whereas [^18^F]FDG showed a similar uptake in infectious and inflammatory lesions, suggesting that [^18^F]FAG could be a promising bacterial infection PET imaging agent [55]. No potential sources of bias were reported for this study.

Recently, the [^18^F]FIAU ability to detect bacterial infection was studied in a mouse model injecting different amounts of wild type and thymidine kinase (TK)-engineered *P. aeruginosa*. Results demonstrated that the TK-engineered *P. aeruginosa* is a useful tool to study bacterial infection in a preclinical setting and [^18^F]FIAU revealed to be an attractive radiopharmaceutical for detecting bacteria and evaluating drug efficacy [61]. Also in this study, the lack of animal origin could have introduced some biases. However, it is also worth noticing (see above) that previous studies with a ^124^I-labelled equivalent fell short when translated into clinical human trials [40]. 

In summary, all ^18^F-labelled radiopharmaceuticals were able to discriminate the presence of Gram-negative bacteria from sterile inflammation or healthy control and, when compared to [^18^F]FDG, they showed better results. The QUADAS analysis revealed that the main source of bias was related to the unspecified origins of the used animals or bacteria.

#### 3.3.3. Copper-64 (^64^Cu)-Labelled Radiopharmaceuticals

Infection caused by *Y. enterocolitica* was induced in C57BL/6 mice to study the ability of [^64^Cu]copper-NODAGA-labelled Yersinia-specific polyclonal antibodies targeting the outer membrane protein YadA in comparison to [^18^F]FDG. Experimental data showed a rapid, sensitive and specific detection of infection by the radiopharmaceutical, whose uptake decreased after blocking [41]. The non-specified origin of animals and bacterial cells might have introduced risks of bias for the QUADAS analysis.

#### 3.3.4. Gallium-68 (^68^Ga)-Labelled Radiopharmaceuticals

A depsidomycin derivative, TBIA101, was radiolabelled with gallium-68 in order to investigate its specificity in BALB/c mice infected with *E. coli* for targeting bacterial lypopolisaccharide (LPS) with PET/CT. Because of suboptimal imaging results, further studies are required to confirm its potential as a radiopharmaceutical to image infection [45]. As a possible bias of the study, the selection of animals was not clarified in the text and the reference standard was not accurately chosen. Moreover, the small number of included animals could lead to difficult evaluation of future applicability.

#### 3.3.5. Gram-Positive and Negative

As patients can normally not be categorized as being infected with either Gram-positive or Gram-negative species when imaging is employed to find occult infectious foci, for a radiopharmaceutical to be sufficiently sensitive to be employed upfront in the diagnostic workup, it should detect both overall groups while still not be taken up in sterile inflammation or cancer cells. Thus, some authors have looked into this by exploring potential radiopharmaceuticals in settings with both Gram-positive and Gram-negative bacteria. 

Ordonez et al. tested three potential radiopharmaceuticals based on an elaborate screening procedure with subsequent in vitro testing: para-aminobenzoic acid (PABA) (with uptake in several different bacterial strains including mycobacteria, *E. coli*, and *S. aureus*), mannitol (with uptake in *S. aureus* and *E. coli*), and sorbitol (with uptake in *E. coli* only). All three molecules were tested in a mouse model of myositis with fluorine-labelled analogs that accumulated significantly more in viable bacteria compared to heat-killed bacteria. However, only [^18^F]FDS is readily available as a true fluorine-based radiopharmaceutical, whereas PABA and mannitol were tested as fluorinated analogs. Thus, although potentially useful, a significant preclinical testing program is necessary before translational evaluations are possible for the two most versatile molecules (PABA and mannitol), whereas [^18^F]FDS is readily available, but has the narrowest usage (*E. coli* only) [39].

Gowrishankar et al. evaluated 6”-[^18^F]-fluoromaltotriose, a second-generation radiopharmaceutical targeting the bacteria specific maltose-maltodextrin transporter unique to bacteria and present in various both Gram-positive and Gram-negative species including *S. aureus*, *E. coli*, and *P. aeruginosa*. Maltotriose was investigated because previous studies of fluoro-labelled maltose and maltohexose showed a less favorable biodistribution profile with poor signal-to-noise ratio especially in the pulmonary organs. 6”-[^18^F]-fluoromaltotriose was tested in several in vitro settings to establish a better biodistribution profile. Subsequently, a myositis mouse model was employed to compare uptake in viable *E. coli* compared to uptake in heat-killed *E. coli*. Furthermore, a *P. aeruginosa* wound model was investigated. The uptake of 6”-[^18^F]-fluoromaltotriose was significantly higher in viable bacteria compared to dead bacteria and normal tissue. The authors also performed a response evaluation study in a single rat with incidental *S. aureus* infection – high specific uptake at the site of infection subsided significantly following relevant antibiotic treatment. Thus, 6”-[^18^F]-fluoromaltotriose has the potential to image several different bacteria species, albeit without the ability to differentiate between them. Results also pointed to less favorable results for bacteria that invade cells, e.g., *M. tuberculosis* [36].

Sellmyer et al. bioengineered a novel fluorine-18 labelled radiopharmaceutical based on the antibiotic trimethoprim ([^18^F]FPTMP) targeting the enzyme dihydrofolate reductase (dhfr) in the DNA synthesis and known to have higher affinity for bacterial dhfr than human dhfr. Promising in vitro results established a favorable biodistribution and significantly higher uptake in viable bacteria compared to heat-killed bacteria. Subsequently, a mouse myositis model was employed to test uptake in *S. aureus*, *E. coli*, and *P. aeruginosa* compared to turpentine-induced sterile inflammation, and implanted mammary carcinoma cells. The authors demonstrated a 3-fold increased target-to-muscle ratio in bacterial infections with *E. coli* and *S. aureus*, whereas no uptake was seen in *P. aeruginosa*, which was also corroborated by lower uptake in *P. aeruginosa* in vitro. No uptake was seen in sterile inflammation or cancer cells. Conversely, when [^18^F]FDG was employed in the same model, increased uptake was seen in all three etiologies. A caveat in the present study is the potential for resistance towards the antibiotic’s mode of action which may render such a radiopharmaceutical useless. Furthermore, when different inoculation amounts were tested, [^18^F]FPTMP was not able to detect infections by less than 10^8^ CFU, suggesting that the number of infection cells play a role [35].

Finally, Ebenhan et al. investigated the Ga-68-labelled antimicrobial peptide TBIA101, a so-called depsipeptide derivative proposedly especially active against multidrug resistant mycobacterium species through interaction with the bacterial cell envelope. Following biodistribution studies, radiopharmaceutical uptake was evaluated in a mouse model with muscle infection (*S. aureus* or *M. tuberculosis*) compared to sterile turpentine-induced inflammation. Significantly increased radiopharmaceutical uptake was seen in *M. tuberculosis* infection (T/NT ratio of 2.6), but none in *S. aureus* infection. Furthermore, a similarly increased uptake with T/NT ratio of 2.6 was also found in sterile inflammation. Thus, these results suggest a lack of bacterial selectivity for TBIA101 and therefore a limited usefulness in specific bacterial imaging [38].

In summary, three radiopharmaceuticals showed to be specific for bacterial infection imaging by using both Gram-positive and Gram-negative bacteria. However, all three demonstrated some caveats regarding the compounds’ applicability; one pointed to less favorable results for bacteria that invade cells, e.g., *M. tuberculosis* [36]; one was based on an antibiotic with the potential for limited use in case of resistance and also results pointing to less efficacy with smaller amounts of infective cells [35]; and one study reported a poor specificity for the studied radiopharmaceutical, limiting its translation to humans [38]. The QUADAS analysis revealed that the main source of bias was related to unspecified origins of animals or bacteria.

### 3.4. Other Pathogens 

Several microbial organisms not readily divisible into Gram-positives and Gram-negatives have also been subjected to more or less specific PET-radiopharmaceuticals other than [^18^F]FDG. Thus, we included 6 studies for the analysis, excluding those that did not meet the inclusion criteria. In total, 5 radiopharmaceuticals and 2 different bacterial strains were considered in the QUADAS analysis.

Invasive pulmonary aspergillosis (IPA) is a life-threatening lung disease caused by the fungus *Aspergillus fumigatus*, a leading cause of invasive fungal infection-related mortality and morbidity in patients with hematological malignancies and bone marrow transplants. Several radiopharmaceuticals labelled with various isotopes have been investigated. Petrik et al. explored different ^68^Ga-labelled radiopharmaceuticals: first, they described the uptake of ^68^Ga-siderophores, [^68^Ga]galliumTAFC and [^68^Ga]galliumFOXE respectively, in invasive pulmonary aspergillosis. Their research showed both [^68^Ga]galliumTAFC and [^68^Ga]galliumFOXE can be used for imaging of IPA with [^68^Ga]galliumFOXE being slightly superior in terms of sensitivity. [^68^Ga]galliumTAFC showed also high in vitro specificity towards *A. fumigatus* compared to other tested micro-organisms and human lung cancer cells. In the second study, the same group also used ^68^Ga-radiolabelled siderophores to visualize IPA in *A. fumigatus*-infected animals with µPET/CT. These small high-affinity chelating compounds are produced by fungi and bacteria to scavenge iron from the host, and by Gram-negative bacterial pathogens as virulence factors. The high metabolic stability, favorable pharmacokinetics with rapid renal excretion and high specific uptake in *A. fumigatus* cultures were confirmed in imaging studies in a rat IPA model that showed high focal uptake in infected lung tissue corresponding to pathological findings seen on CT. High specific uptake in *A. fumigatus* cultures is somewhat reduced under conditions of iron overload, which is what patients who acquire fungal infections typically suffer from [53,60]. Rolle et al. used their own newly developed [^64^Cu]copper-DOTA-labelled *A. fumigatus*-specific monoclonal antibody (mAb), JF5, in neutrophil-depleted *A. fumigatus*-infected mice, which allowed specific localization of lung infection when combined with PET/MR. This radiopharmaceutical distinguished IPA from bacterial lung infections and, in contrast to [^18^F]FDG-PET, discriminated IPA from a general increase in metabolic activity associated with lung inflammation [43]. Severin et al. showed, that [^89^Zr]zirconium-oxalate can exhibit substantial tumor accumulation as well as significant accumulation in *Aspergillus* infected lungs as compared to healthy lungs [47]. They suggest control experiments mapping the biodistribution of free zirconium-89 in any preclinical study employing zirconium-89 where bone uptake is observed.

*Mycobacterium tuberculosis* was studied in the same model of infection by two groups that used two different radiopharmaceuticals. Weinstein et al. radiolabelled an analogue of isoniazid (INH) with fluorine-18 to verify its ability as a radiopharmaceutical for PET imaging of *M. tuberculosis*. They reported an increasing uptake of [^18^F]INH in tuberculosis (TB) lesions of infected animals over time in comparison to non-infected ones, proposing [^18^F]INH as a promising agent for identifying TB infection. [^18^F]FDG on the other hand was not able to discriminate between infection and inflammation [52]. The unclear origin of bacterial cells and reference standard might have introduced biases in the study.

More recently, *M. tuberculosis* infection was studied in a preclinical model using an analogue of the first-line tuberculosis drug, the pyrazinamide (PZA), radiolabelled with fluorine-18. Results showed a successful radiolabelling procedure of the compound, but also a comparable distribution in TBC necrotic areas between infected and uninfected mice, thus limiting its potential use [37]. No sources of bias were reported for this study.

## 4. Discussion

Bacterial infections are still one of the main causes of mortality worldwide. Many efforts have been made to develop new specific radiopharmaceuticals for PET or SPECT imaging to improve the overall diagnostic accuracy [69,70,71]. The continuous development of bacteria-specific imaging agents to discriminate between septic and sterile inflammation testifies the need for a highly accurate tool for infection diagnosis and therapy follow-up.

In this setting, we have previously published a meta-analysis of SPECT radiopharmaceuticals for imaging bacteria [3] and we now performed a perceptive analysis of the literature concerning PET radiopharmaceuticals for bacterial imaging in preclinical studies. Considering the heterogeneity of included manuscripts, we divided them into four groups in relation to the bacterial strain to which the described radiopharmaceutical is directed. Each paper was analysed following the QUADAS guidelines that verify the risk of bias, quality of studies and applicability, adjusting these parameters to our papers (Appendix A) 

Most of the authors that used Gram-positive bacteria reported the suitability of the tested radiopharmaceuticals for bacterial imaging in animal models but also in patients. In particular, the ubiquicidin peptide fragments, both UBI-31-38 and UBI-29-41, allowed to image foci of *S. aureus* infection in preclinical models with high T/NT ratios. Recently, UBI fragments were tested in humans and revealed to be safe and non-toxic, although further studies are needed to confirm preliminary results [28,32,44,51]. Other radiopharmaceuticals, like [^124^I]FIAU, were less encouraging when translated to patients despite initial promising results in animals [40]. Other groups showed high specificity of radiopharmaceuticals in the discrimination between infection and sterile inflammation, thus considering them as potential agents for bacterial imaging in humans [31,34,46]. Worth of mention is the case of radiolabelled ciprofloxacin: discordant data when labelled with technetium-99m, unsuccessful when labelled with fluorine-18 [29,59], and highly specific when labelled with gallium-68 [58]. These controversial results are probably due to the difference in isotopes used and the chemistry to label with different isotopes that may influence the biological behaviour of radiolabelled-ciprofloxacin. 

In the majority of papers on Gram-negative bacteria, promising results were reported by using radiolabelled sugars (sorbitol, maltose, maltohexaose and FAG). Indeed, several groups showed high specificity of [^18^F]FDS binding to *E. coli* or *K. pneumoniae* [33,42,48]. Also, other sugars such as FMH, 6′’-[^18^F]-fluoromaltose and [^18^F]FAG revealed to be sensitive and specific radiopharmaceuticals for detection of *E. coli* [49,50,55].

Regarding the QUADAS analysis for papers that used Gram-positive bacteria, the most frequent sources of bias were related to the origin of animals (about 50% of total articles) and the index test (about 50% of total articles), particularly the origin of bacterial cells and the not standardized infection model. Only two papers did not show risks of bias and applicability concerns in each section [34,62]. 

QUADAS analysis of papers on Gram-negative bacteria, only four of them showed a low risk of bias and low concerns about applicability through an appropriate experimental procedure, avoiding the introduction of biases [30,33,49,55]. The most frequent source of bias was related to animal selection and index test (the origin of bacterial cells). Indeed, it is extremely important to clarify the origin of animals/bacteria with the aim to perform reproducible and standard methods.

Some group used both Gram-positive and Gram-negative bacteria or a combination of the with *M. tuberculosis* to induce infection in preclinical models. Out of four papers, three reported good specificity of radiopharmaceuticals for bacterial infection imaging with future applications in clinics [35,36,39]. On the other hand, [^68^Ga]gallium-DOTA-TBIA101 was found to be non-specific when the infection was induced by using *S. aureus* or *M. tuberculosis* [38], while it was considered a promising agent for *E. coli* bacterial infection [45]. Indeed, [^68^Ga]gallium-DOTA-TBIA101 requires further studies to confirm its specificity for bacterial imaging and for which bacterial strain.

Concerning QUADAS analysis for these papers, the most frequent sources of bias were related to the origin of animals or bacterial cells (50% of total articles). Only one did not show risks of bias and applicability concerns in each section [36].

Finally, all groups that tested radiopharmaceuticals to image *M. tuberculosis* and *A. fumigatus* infections in preclinical models, reported high sensitivity and specificity to localize infectious foci, making them promising agents for infection imaging in humans too [37,43,47,52,53,60]. 

Regarding QUADAS analysis for papers that used other bacteria (*M. tuberculosis*, *A. fumigatus*), five articles out of six (83.3%) presented low risks of bias.

Overall results suggest that, despite promising initial findings, more studies are needed to further confirm the diagnostic use of any of the described radiopharmaceutical. Better study design with standardized and reproducible infection models, an appropriated reference standard and a pertinent study flow, are needed. 

Despite many overall promising results, no radiopharmaceutical has been introduced into clinical practice yet. This is probably because specific and non-specific mechanisms coexist at the infection site leading to a bias in the analysis of the results. 

The most critical issue that we observed in all studies is related to the infection model, controls and general experimental setting such as the amount of bacterial cells, the radiopharmaceutical dose, the imaging time after radiopharmaceutical injection, the different time interval between bacteria injection, the radiopharmaceutical administration route, and the imaging time points. In addition, none of the reviewed papers take into consideration possible bacterial mutations that may occur during an infection, thus changing their properties. These mutations may alter the radiopharmaceutical uptake that could lead to a false negative scan (i.e., during antibiotic therapy or during epidemic infections). 

The Teflon tissue cage model is a good example of what could be considered a standardised reproducible animal model to study bacterial infections [65,72]. Indeed, once implanted into the back of the mouse, it provides several advantages such as a localized infection, an accurate determination of bacterial mass and radiopharmaceutical concentration over time or the study of biofilm formation [73]. 

The lack of common experimental models and procedures did not allow us to compare the results obtained using different radiopharmaceuticals and prediction of results in humans is difficult. 

Furthermore, in the light of data published so far, it emerges that there is the lack of knowledge whether it is possible to develop an all-purpose radiopharmaceutical to image all bacterial strains. Nowadays this remain an open goal, difficult to achieve, but, at the same time, crucial for the management, treatment and follow-up of patients with suspected bacterial infections.

## 5. Conclusions

This review highlights that several new PET radiopharmaceuticals for bacterial imaging have been developed with potential to be translated to humans for PET/CT imaging of infection. Despite initial promising results, however, many candidate compounds remain confined in a pre-clinical setting due to the presence of biases that limits their impact. When designing new studies, standardized protocols and models could ensure that time, funds, and research efforts are put to the best possible use. Indeed, the most frequent sources of bias found in selected articles were related to animal selection and index test and showed the lack of standardization with current infection models and experimental settings. Thus, standardized protocols and consensus guidelines regarding animal models of infection are needed, preferably written by a joint technical committee.

## Figures and Tables

**Figure 1 jcm-08-00197-f001:**
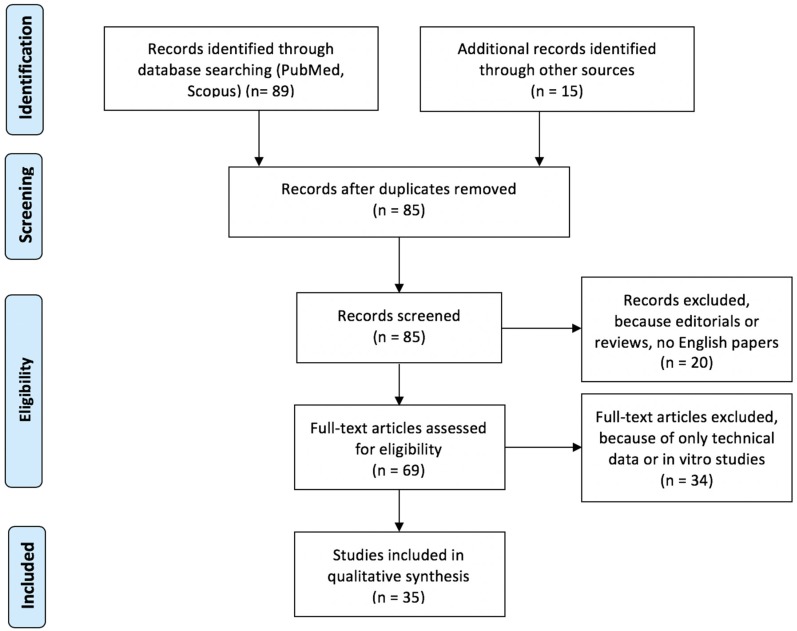
Preferred Reporting Items for Systematic Reviews and Meta-analyses (PRISMA) flowchart of included papers.

**Table 1 jcm-08-00197-t001:** Summary table of papers included in the systematic review.

First Author [ref]	Pathogen	Compound	Radiolabeling Method	Isotope	Specific Activity	Stability	Model of Study	Metabolic Route	Amount (CFU) and Infection site	Max Target-to-non Target (T/NT) Ratio	Control Experiment	Other	Comments by Authors
Bhatt J [28]	*S. aureus*	UBI-31-38	1,4,7-triazacyclononane-N,N′,N″-triacetic acid (NOTA) chelator	^68^Ga	2.1 × 10^6^ MBq/mmol	both in saline and serum up to 2 h	BALB/c mice	kidneys	10^7^, right thigh	3.24 ± 0.7	10^7^ heat killed, left thigh	imaging in 2 patients and 1 negative control	good localization of infection site, high T/NT ratio and promising results in humans
Nielsen KM [29]	*S. aureus*	K-A9 peptide	1,4,7,10-Tetraazacyclododecane-1,4,7,10-tetraacetic acid (DOTA) chelator	^68^Ga	1.4 × 10^4^ MBq/mmol	up to 2 h	C57BL/6 mice	kidneys	5 × 10^7^	1.89 ± 0.88	turpentine oil	comparison with 2-deoxy-2[^18^F]fluoro-D-glucose (FDG)	no in vivo selectivity
Mutch CA [30]	*E. coli*	Para-Aminobenzoic Acid (PABA)	argon carrier gas	^11^C	820 ± 258 mCi/μmol	-	CBA/J mice	kidneys	10^8^, left shoulder	2.8 vs. 1 (region of interest)	10^8^ killed, right shoulder	-	attractive candidate for imaging living bacteria in humans.
Takemiya K [31]	*S. aureus*	Maltohexaose (FHM)	nucleophilic fluorination	^18^F	-	-	Sprague-Dawley rats	feces, urine	2.9 × 10^8^, stainless steel implant, back	-	turpentine oil	negative controls comparison with FDG	FHM is better than FDG in differentiating non-infection inflammation from infection
Ebenhan T [32]	only in vitro test, in vivo biodistribution in non-human primates (NHP) and humans	UBI-29-41	NOTA chelator	^68^Ga	13.8 ± 1.9 GBq/mmol	-	vervet monkeys (NHPs)	liver, kidneys	-	3.3 ± 0.3 maximum standard uptake value (SUV_max_) at 1 h	-	human studies (2 healthy and 3 patients)	non-toxic, safe compound, identify infectious foci in humans; need further studies
Li J [33]	*K. pneumoniae*	Sorbitol (FDS)	nucleophilic fluorination	^18^F	-	-	C57BL/6 mice	kidneys	10^5^ live, lungs	8.5 (FDS) and 1.7 (FDG)	10^8^ killed, lungs	comparison with FDG	FDS is better than FDG to detect lung infection from inflammation
Pickett JE [34]	Gram-positive	SAC55 (anti-LTA)	Deferoxamine (DFO) chelator	^89^Zr	9.439–12.210 (1.7–2.2 mCi/g)	-	C57BL/6 mice	-	4.3 × 10^4^, femur, inoculated implant	-	1.4 × 10^3^, sterile implant	comparison with FDG, NaF, ^89^Zr-IgG (control)	potential differentiation between infection and sterile inflammation
Sellmyer MA [35]	*E. coli* and *P. aeruginosa* *S. aureus*	trimethoprim	nucleophilic fluorination	^18^F	5–15 × 10^6^ mCi/mmol	-	BALB/c mice and NHP (rhesus monkey) (biodistribution)	kidneys, liver	10^6^ hindlimb, 10^7^ forelimb, 10^8^ ear pinna	2.7 vs. 1.3 vs. 1 (infection vs. inflammation vs. tumor)	10^6^, 10^7^, 10^8^ killed, contralateral left	comparison with inflammation and cancer performing FDG too	specific for infection imaging
Gowrishankar G [36]	Gram-positive Gram-negative	maltotriose	nucleophilic fluorination	^18^F	-	-	nude mice (*E. coli*), CD1 mice (*P. aeruginosa*, *L. monocytogenes*), BALB/c mice (LPS)	kidneys	10^8^ *E. coli*, 10^6^ *P. aeruginosa*, 2 × 10^5^ *L. monocytogenes*	-	Lipopolysaccharide (LPS) inflammation, 10^6,7,8^ heat killed *E. coli*	nude rat infected with aureus to monitor antibiotic therapy efficacy	able to image bacterial infections in animals with future applications in clinics
Zhang Z [37]	*M. tuberculosis*	pyrazinamide analog	halogen exchange reaction	^18^F	2.6 × 10^6^ kBq/µmol	60% labelling efficiency (LE) at 90′ in livers	C3HeB/FeJ	bone, lungs	lungs	-	uninfected animals	-	successful radio-synthesis, higher uptake by infected lungs
Ebenhan T [38]	*M. tuberculosis S. aureus*	TBIA101 (depsidomycin derivative)	DOTA chelator	^68^Ga	13 ± 6 GBq/µmol	-	New Zealand white rabbits	kidneys	10^8^, thigh muscle	2.8 ± 0.16 and 2 ± 0.31 at 1 h for *M. tubercolosis* thigh (MTB)/triceps and MTB/thigh	turpentine oil, contralateral or *M. tubercolosis*, contralateral	-	imaging inflammation with turpentine oil, but not infection. Non-specific
Ordonez AA [39]	*E. coli* and *P. aeruginosa* *S. aureus* *M. tubercolosis*	PABA, mannitol, FDS	-	^3^H for PABA and mannitol, ^18^F for FDS	-	-	CBA/J mice	kidneys	10^6^ *S. aureus* (PABA), 10^7^ *E. coli* (mannitol and FDS), right thigh	-	heat killed bacteria, left thigh	-	have significant potential for clinical translation for the rapid diagnosis of bacterial infections
Zhang XM [40]	*S. aureus*	Fialuridine (FIAU)	-	^124^I	999–1295 GBq/µmol	-	patients suspected of prosthetic joint infection (PJI)	kidneys	-	-	-	-	^124^I-FIAU is well tolerated but of limited value for detection of PJI due to low image quality and low specificity
Wiehr S [41]	*Y. enterocolitica*	polyclonal monoclonal Antibody (mAb) vs. *Yersinia* adhesin A (YadA)	1,4,7-triazacyclononane,1-glutaric acid-4,7-acetic acid (NODAGA) chelator	^64^Cu	650–730 MBq/mg	90% up to 48 h both in saline and serum	C57BL/6 mice	spleen	10^3^–10^4^, i.v.	-	PBS and blocking experiments	comparison with FDG	rapid, sensitive and specific imaging probe
Yao S [42]	*E. coli*	FDS	nucleophilic fluorination	^18^F	29.6 ± 6.5 GBq/μmol	95% up to 2 h in serum	C57BL/6 mice	kidneys	10^5^–10^7^, right thigh	2.05 ± 0.07 and 13.0 ± 1.35, inflammation vs. infection	turpentine oil, left thigh	comparison with FDG, human studies	promising probe for diagnosis and monitoring therapy
Rolle AM [43]	*A. fumigatus*	JF5 mAb	DOTA chelator	^64^Cu	-	up to 48 h in serum	neutropenic C57BL/6 mice	lungs, liver, spleen	4 × 10^6^, lungs	11.9 ± 1.3, 7.7 ± 2, 8.1 ± 0.6, 8.1 ± 1.5 (%ID/cc for *A. fumigatus*, PBS, *S. pneumoniae*, *Y. enterocolitica*)	10^6^ *S. pneumoniae*, 5 × 10^4^ *Y. Enterocolitica* or PBS	comparison with FDG and blocking experiments	mAb localized aspergillus infection. FDG is non-specific for imaging infection
Vilche M [44]	*S. aureus*	UBI-29-41	NOTA chelator	^68^Ga	0.55 × 10^6^	3 h	*M. musculus* Swiss mice	kidneys	1.2 × 10^8^, left thigh (measured in vitro 5 × 10^8^–1.2 × 10^9^ and as low as 2.5 × 10^4^)	infection 5 (in vitro) 1 h inflammation 1.6 (in vitro), 1 h 4.0 (PET), 1 h Non-significance inflammation/normal	(1) 1.2 × 10^8^ heat killed, left thig (2) healthy mice	-	clearly observed different uptake in infection vs. inflammation positively correlated with degree of infection
Mokaleng BB [45]	*E. coli*	TBIA101 (depsidomycin derivative)	DOTA chelator	^68^Ga	12.4 ± 6 GBq/μmol	>97% up to 3 h in serum	BALB/c mice	liver, kidneys	5 × 10^8^, right thigh	3 ± 0.8 (infection) vs. 2.3 ± 0.6 (inflammation)	healthy muscle	-	promising agent but need further studies
Mills B [46]	*S. aureus*	FDG-6-P	-	^18^F	injected activity 8–11 MBq (FDG and FDG-6-P)	3 h	BALB/c mice	kidneys	1 × 10^9^, via catheter in right flank	~3.0 (FDG-6-P) ~1.0 (FDG)	saline, via catheter in right flank	-	in vitro validated method, but very different behavior in vivo with high background, FDG-6-P accumulation had higher T/NT ratio than FDG = potential to differentiate infection from inflammation
Severin GW [47]	*A. fumigatus*	free ^89^Zr (oxalate form)	direct labelling	^89^Zr	20–35 GBq/μmol	-	neutropenic C57BL/6 mice	kidneys	4 × 10^6^, intratracheal	6.3 vs. 3.8	healthy neutropenic mice	-	injection of free ^89^Zr (oxalate at pH < 7), routinely be performed as a control experiment
Weinstein EA [48]	Enterobacteriaceae	FDS	nucleophilic fluorination	^18^F	-	-	CBA/J mice	intestine, kidneys	10^7^, right thigh	-	10^7^ heat killed, left thigh; mixed infection with aureus; brain tumor	comparison with FDG	diagnostic tool for imaging infections due to Enterobacteriaceae
Ning X [49]	*E. coli*	maltohexaose	nucleophilic fluorination	^18^F	-	-	rats	spleen, kidneys	10^5–7^, left triceps	8.5 at 70′	PBS or 10^9^ killed, right triceps	comparison with FDG	sensitive, specific radiopharmaceutical that can identify drug resistance
Gowrishankar G [50]	*E. coli*	maltose	nucleophilic fluorination	^18^F	-	-	nude mice and BALB/c mice	kidneys	5 × 10^7^, right thigh	4.2 at 3 h	10^8^ heat killed, contralateral and turpentine oil in BALB/c	-	promising new radiopharmaceutical for bacterial infection imaging
Ebenhan T [51]	*S. aureus*	UBI-29-41	NOTA chelator	^68^Ga	injected activity 27 ± 11/29 ± 15/29 ± 15 MBq (normal controls/thighs /lung)	4 h	New Zealand white rabbits	kidneys	2 × 10^8^, right thigh	4.35 ± 0.85 (infection/normal) 3.54 ± 0.86 (infected/inflammation)	turpentine oil, left thigh + normal controls + lung inflammation (asthma)	-	^68^Ga-UBI strongly localized in infection and only minimally in inflammation. No uptake in lung inflammation either
Weinstein EA [52]	*M. tuberculosis*	isonicotinic acid (INH)	nucleophilic fluorination	^18^F	7.4 to 11.1 MBq/µmol	-	C3HeB/FeJ mice and BALB/c mice	kidneys, liver	log10(6.4 ± 0.3) CFU, lungs	1.67 ± 0.04	non-infected controls	comparison with FDG	rapid, non-invasive approach to localize infectious foci
Petrik M [53]	*A. fumigatus*	siderophores: triacetylfusarinine (TAFC) and ferrioxamineE (FOXE)	direct labelling	^68^Ga	9.2 × 10^4^ and 3.4 × 10^3^ GBq/mmol, TAFC and FOXE	-	BALB/c mice for biodistribution Lewis rat	kidneys	10^5^–10^9^ conidia/ml, lungs	5.81 ± 6.05 vs. 0.78 ± 0.75 (TAFC), 6.64 ± 2.91 vs. 1.0 ± 0.81 (FOXE) as SUV	non-infected controls	-	very promising agents for detection of infection with high sensitivity
Panizzi P [54]	*S. aureus*	prothrombin (ProT)	Diethylene triamine pentaacetic acid (DTPA)	^64^Cu	Injected doses 0.92–1.62 mCi	-	C57BL/6 mice	-	1 × 10^6^/µL, aortic valve	Only visual assessment	Non-infected controls	-	non-invasive detection of *S. aureus* induces endocarditis is feasible with an engineered analog of prothrombin
Martínez ME [55]	*E. coli*	fluoroacetamido-D-glucopyranose (FAG)	microwave irradation	^18^F	18.09 ± 2.9 GBq/μmol	-	Sprague-Dawley rats	kidneys	10^7^, right thigh	0.54 ± 0.21 vs. 0.19 ± 0.07	turpentine oil	comparison with FDG	promising infection agent
Kumar V [56]	*S. aureus*	Transferrin (TF)	apo	^68^Ga	Injected dose:10–15 MBq	6 h	Wistar rats	-	5 × 10^5^, right thigh	2.2-7.5	-	-	^68^Ga-TF is capable of detecting *S. aureus* infection
Diaz Jr. LA [57]	*S. aureus*	FIAU	-	^124^I	318 × 10^6^ (8.596–13.979 Ci/mmol)	-	patients suspected of musculoskeletal infections (n=8)	kidneys	various locations	-	1 control patient	surgery as gold standard	promising new method, but preliminary data
Satpati D [58]	*S. aureus*	ciprofloxacin	DOTA and NOTA chelators	^68^Ga	6.2 ± 0.4 MBq/nmol	both in saline and serum up to 4 h	Wistar rats	kidneys	5 × 10^7^, right thigh	1.5 for DOTA, 5 for NOTA	turpentine oil, left thigh	-	need further investigations
Langer O [59]	Gram-positive Gram-negative	ciprofloxacin	nucleophilic fluorination	^18^F	342 ± 94 MBq/µmol	-	4 patients with proven soft tissue infections	-	-	at peak uptake SUV = 5.5		-	not suitable as specific agent
Petrik M [60]	*A. fumigatus*	TAFC and FOXE	direct labelling	^68^Ga	3.4 × 10^6^ (FOXE) MBq/mmol	80% (TAFC) and 90% (FOXE) at 2 h in serum	rats	lungs	n.a./left calf	-	turpentine oil and 10^9^ CFU of *S. aureus* (i.m.)	comparison with FDG	high selective accumulation in infected lungs
Rajamani S [61]	engineered *E. coli* and *P. aeruginosa*	FIAU	nucleophilic fluorination	^18^F	-	-	CD1 mice and BALB/c mice	gastrointestinal tract	10^5^, 10^7^, 10^9^, thigh	-	*P. aeruginosa* infection	therapy with ciprofloxacin	engineered pathogens for evaluating experimental therapeutics
Zhang Z [62]	*S. aureus*	PABA	Radio-synthesis box	^18^F	240.5 ± 77.7 GBq/µmol	-	Sprague-Dawley rats	kidneys	10^8^, left triceps	5 at 1 h	heat killed, contralateral	comparison with FDG	novel, non-invasive diagnostic tool for detecting, localizing, and monitoring *S. aureus* infections

**Table 2 jcm-08-00197-t002:** Summary of QUADAS analysis.

	Risk of Bias				Applicability Concerns		
First Author [ref]	Animal Selection	Index Test	Reference Standard	Flow and Timing	Animal Selection	Index Test	Reference Standard
**Gram-positive**							
Bhatt J [28]							
Nielsen KM [29]							
Takemiya K [31]							
Ebenhan T [32]							
Pickett JE [34]							
Zhang XM [40]							
Vilche M [44]							
Mills B [46]							
Ebenhan T [51]							
Panizzi P [54]							
Kumar V [56]							
Diaz Jr. LA [57]							
Satpati D [58]							
Langer O [59]							
Zhang Z [62]							
**Gram-Negative**							
Mutch CA [30]							
Li J [33]							
Wierhr S [41]							
Yao S [42]							
Mokaleng BB [45]							
Weinstein EA [48]							
Ning X [49]							
Gowrishankar G [50]							
Martìnez ME [55]							
Rajamani S [61]							
**Gram-Positive and Negative**							
Sellmyer MA [35]							
Gowrishankar G [36]							
Ebenhan T [38]							
Ordonez AA [39]							
**Others**							
Zhang Z [37]							
Rolle AM [43]							
Severin GW [47]							
Weinstein EA [52]							
Petrik M [53]							
Petrik M [60]							

Green: “+”, Red: “−”, Yellow: “?”.

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
