# Peer review of "PET Radiopharmaceuticals for Specific Bacteria Imaging: A Systematic Review"

_jcm, 2019, doi:10.3390/jcm8020197_

Reviewer 1 Report

This manuscript is well structured and cover, to the best knowledge of the reviewer, the topics of PET and bacteria imaging, which is of clinical interest and the review is thus of value.

The manuscript should make some editorial correction regarding nomenclature for radiopharmaceutical.NOMENCLATURE FOR RADIOLABELLED COMPOUNDS and EDITORIAL COMMENTS

[18F]-FDG  change to:  [18F]FDG

the same change (removing hyphen) should be done for all other compound names e.g

[18F]-FDG-6-phosphate,

all compound names, such as 68Ga-apo-transferrin, 68Ga-NOTA-UBI-29-41 etc should use Square brackets,  [68Ga]ap-

AND if fully correct it should be for all metal chelates (gallium, Tc and zirconium): for example, [68Ga]gallium-NOTA-UBI-29-41   (NOTA-UBI-29-41 doesnot contain gallium and [68Ga] means that a minor (trace amounts) fraction of all Ga atoms are gallium-68) and the same applies for 89ZrSAC55 [[89Zr]zirconiumSAC55, unless SAC55 do contain zorconium. There are many other examples of these types of incorrect nomenclature, and there is an ongoing effort in the radiopharmaceutical scientific community to work towards a consistent and correct nomenclature in scientific literature.

68GaCl3 = [68Ga]GaCl3

in text for example line 150  "Interestingly, 89Zr was...." should be " Interestingly, zirconium-89 was..." same applies for all radionuclides.

line 219 Carbonium (11C)  should be  Carbon (11C)

68Ga-radiopharmaceutical, not a specified compound, is correct without square brackets and [124I]-labelled;  should thus be 124I-labelled

line 306: "...as pseudo-fluorinated analogs. This does not make sense, it can only be "...fluorinated analogs of x and Y."

line 325: fluoro-labelled radiopharmaceuticals  should be fluorine-18 labelled .....

line 307 enzyme dhfr, plese explain what dhfr stands for

page 25 line 9-11. "Worth of mention is the case of radiolabelled ciprofloxacin: discordant data when labelled with 99mTc, unsuccesful when labelled with 18F, and highly specific when labelled with 68Ga."

The statement is of course correct but not very strange if considering, that all three labelled compounds are different chemicals entities, with of course different biological characteristics, none of them are "ciprofloxacin" just an analog of.

Author Response

The manuscript should make some editorial correction regarding nomenclature for radiopharmaceutical.

 [18F]-FDG change to: [18F]FDG  Done

 [18F]-FDG-6-phosphate,Done

 [68Ga]ap-Done

 [68Ga]gallium-NOTA-UBI-29-Done

68GaCl3 = [68Ga]GaCl3Done

"Interestingly, 89Zr was...." should be "Interestingly, zirconium-89 was..." same applies for all radionuclides.Done

Carbonium (11C) should be Carbon (11C)Done

68Ga-radiopharmaceutical, not a specified compound, is correct without square brackets and [124I]-labelled; should thus be 124I-labelledDone

"...as pseudo-fluorinated analogs. This does not make sense, it can only be "...fluorinated analogs of x and Y."Done

fluoro-labelled radiopharmaceuticals should be fluorine-18 labelled .....Done

enzyme dhfr, please explain what dhfr stands for Done

"Worth of mention is the case of radiolabelled ciprofloxacin: discordant data when labelled with 99mTc, unsuccessful when labelled with 18F, and highly specific when labelled with 68Ga."

The statement is of course correct but not very strange if considering, that all three labelled compounds are different chemicals entities, with of course different biological characteristics, none of them are "ciprofloxacin" just an analog of. Added a statement explaining reasons of different published results.

Reviewer 2 Report

The manuscript gives a systematic review on imaging bacterial infections using radiopharmaceuticals using PET isotopes. Though sources of studies have discrepancy and biased, the obtained data is well classified by authors wherever possible.

Reviewer questions for knowledge:

1) Bacterial imaging is always a challenging subject; Bacteria can mutate and change its properties in vivo. Is there a possibility of imaging an infection can be done based on bacterial population, no matter whatever the source of bacteria? Taking consideration of typical bacterial properties.

2) As authors discussed about various PET reagents, does half-life of agent matters for good resolution in imaging, taking consideration of bacterial lifetime as<12 hours (or more) approximately?

3) mAb imaging is not right idea as bacteria life time is an average<12 hours or more and mAb excretion time more than that. Pre-targeting is  better option, if researcher  want to do imaging with mAb to overcome the background. (What do you think?)

Please give an opinion to above questions raised.

Minor criticism

1)  Authors should expand all abbreviations, if it is used first time in the manuscript. Correct it.

2)  As the review on imaging, it is expected to show couple of images like comparing FDS and FDG or whatever possible.

Author Response

1) Bacterial imaging is always a challenging subject; Bacteria can mutate and change its properties in vivo. Is there a possibility of imaging an infection can be done based on bacterial population, no matter whatever the source of bacteria? Taking consideration of typical bacterial properties.

Thank you for raising this interesting point. 

Indeed, bacterial mutation is a very relevant problem. Most reviewed manuscript concentrate in developing a bacteria-specific radiopharmaceutical that could lead to a false negative scan if bacteria mutate (i.e. during antibiotic therapy or during epidemic infections). None of the cited papers payed attention to this issue, but indeed bacterial mutations might influence the radiopharmaceutical uptake. This comment has been added in the discussion.

2) As authors discussed about various PET reagents, does half-life of agent matters for good resolution in imaging, taking consideration of bacterial lifetime as<12 hours (or more) approximately?

Indeed the number of bacteria in an infected lesion may change over time. It may decrease due to host response or increase due to bacteria exponential growth. However, as far as bacteria are present a radiopharmaceutical should be able to image them, regardless to their lifetime. Anyhow, none of the reviewed papers did correlate the in vivo uptake of radiopharmaceutical with exact number of bacteria at time of imaging. Such information may be difficult to obtain. 

3) mAb imaging is not right idea as bacteria life time is an average<12 hours or more and mAb excretion time more than that. Pre-targeting is better option, if researcher want to do imaging with mAb to overcome the background. (What do you think?)

Please give an opinion to above questions raised.

Finally, mAbs do not necessarily follow the bacterial growth kinetic as observed for tumor imaging studies too. The presence of the target is more important than its growing kinetics. Nevertheless, pre-targeting could be an option.

Minor criticism

1)  Authors should expand all abbreviations, if it is used first time in the manuscript. Correct it. Done

2)  As the review on imaging, it is expected to show couple of images like comparing FDS and FDG or whatever possible.

Thank you for the suggestion, but being this a systematic review of all manuscript published with PET radiopharmaceuticals for bacteria imaging, we should show an image of each tracer or none. We decided for no images.